# The Effect of Nudging on Compliance with Individual Prevention Measures against COVID-19: An Online Experiment on Greek University Students

**DOI:** 10.3390/ijerph21010031

**Published:** 2023-12-25

**Authors:** Ioannis Emmanouil, Manolis Diamantis, Dimitris Niakas, Vassilis Aletras

**Affiliations:** 1Department of Business Administration, University of Macedonia, 156 Egnatia St., 54636 Thessaloniki, Greece; mhm20032@uom.edu.gr; 2School of Social Sciences, Hellenic Open University, 18 Aristotelous St., 26335 Patra, Greece; std139975@ac.eap.gr (M.D.); dniakas@med.uoa.gr (D.N.); 3Medical School, National and Kapodistrian University of Athens, 4 George St., Kanigos Square, 10677 Athens, Greece

**Keywords:** COVID-19, nudge, framing, preventive behavior, behavioral economics

## Abstract

Nudging has often been suggested as a means to promote health care efficiency and effectiveness by influencing behavior without restricting choice; its usefulness, however, has not been adequately assessed. We examined the effect of an altruistically framed awareness message about the novel coronavirus on the intention to comply with individual prevention measures against infection. A total of 425 Greek postgraduate students, which were randomly assigned into a treatment group and a control group, filled out a questionnaire on compliance and future intention to comply with six preventive measures. The results indicate that the altruistic message did not manage to influence the intention to comply. Moreover, compliance was positively associated with risk perception, whereas women showed both higher compliance and risk perception than men. Vulnerability to the novel coronavirus and a positive vaccination status against it were accompanied by a greater perception of risk, while one’s personal history of COVID-19 was associated with a lower intention to comply, lower risk perception, and higher health risk preferences. We conclude that nudging interventions should be evaluated before being adopted in practice, taking into account timing, target groups, and means of communication.

## 1. Introduction

This paper aims to study the role of the altruistic framing of messages in dealing with the COVID-19 pandemic. Framing constitutes a tool of behavioral economics, a field that may have useful applications in influencing health-related behaviors, thus assisting governmental policies that focus on modifying individual behavior for the common good [1]. In essence, framing is a “nudge”, which is “any aspect of the choice architecture that alters people’s behavior in a predictable way without forbidding any options or significantly changing their economic incentives” [2] (p. 6).

Given the severity of the recent SARS-CoV-2 pandemic, and the importance of complying with individual prevention measures [3], the present work focuses on nudging and its effect on compliance. The previous experience of Eastern countries with SARS contributed significantly to their success in limiting the pandemic spread, as they immediately adapted to increased personal hygiene, the use of masks, and social distancing [4]. Such practices not only protect the user, but also society in general via positive externalities. Therefore, they could be seen either as self-centered or prosocial, and the motivations behind them can be seen as individualistic or altruistic [5]. Although classical economic theory purports that individuals make decisions that maximize their own utility, there is empirical evidence that altruism has great motivational power [6]. 

In health care, both types of motivation, individualistic and altruistic, can encourage vaccination decisions [7,8,9]. Research evidence, however, on their relative effectiveness yields divergent results in different settings. For example, the findings of Isler et al. [10] are indicative of the superiority of the individualistic frame in reinforcing influenza vaccination intentions. On the contrary, a study of young men’s attitudes towards the human papilloma virus vaccine [11] has found that the simultaneous emphasis on both social and personal benefit was accompanied by a greater acceptance of the vaccine. Also, an experiment [12] showed that healthcare professionals significantly increased their hand hygiene when they were exposed to a message reminding them of patients’ benefits. 

Although it is not clear which type of message might be more appropriate in general, an altruistic framing might in fact be preferable in studies that focus on younger individuals. This is because younger people are generally less severely affected by SARS-CoV-2 than the elderly [13]. Therefore, they are less afraid of becoming infected and might not promote compliance with preventive measures with an individualistic message. Prior research [14] has also found that messages trying to induce behavioral change in younger citizens should not focus on making threats to older adults. In line with this finding are the results of an online experiment in Japan [15], which showed that an “altruistic message” that focused on protecting people belonging to their close-family circle reinforced both behavioral intention and actual change in behavior. The evidence, however, is not conclusive in all settings. A study [5] initially demonstrated the superiority of the intervention focusing on social benefit, but at the follow-up, it was found that all interventions were equally effective in terms of the intention to comply. It was also found [16] that framing, with an emphasis on the effects of non-compliance on the subjects themselves and their families, reinforced intentions to stay at home, but did not alter subsequent actual behavior.

Last but not least, people with developed prosociality are more likely to keep social distance, stay home when they are sick, and buy masks [17]. However, there are still conflicting findings, such as those [18] that indicate an insignificant effect from the application of prosocial motivation on the intention to maintain social distance.

Based on the above, our first research hypothesis is formulated as follows:

**Hypothesis** **1.**
*Exposure to an altruistic nudging message has a positive effect on the compliance behavior of individuals.*


Moreover, risk perception may influence compliance with preventive measures and is itself a subjective psychological structure affected by a number of factors, such as cognitive, emotional, social, and cultural factors. Hence, it can differ not only between individuals but also between different societies [19,20]. A study of ten countries in Europe, America, and Asia [21] showed that the COVID-19 risk perception is strongly correlated with the self-reported degree of adopting preventive measures. 

Accordingly, the second research hypothesis is the following:

**Hypothesis** **2.**
*Compliance is associated with risk perception.*


The above literature indicates that there is rather limited and inconsistent evidence regarding the effectiveness of nudging during the pandemic. The contribution of our study lies in the fact that it was conducted in the middle of the COVID-19 pandemic in contrast to prior research that was carried out during its initial phase. It also provides evidence from a European country that has not been studied before. Findings from studies such as ours can be used by policy makers to identify cost-effective nudge strategies that will preserve population health during pandemics.

## 2. Materials and Methods

This study was approved by the Health Care Management (DMY) Programme of Study Committee (Protocol Code 3/20-12-2021), acting as a Committee for Research Ethics of the University. The research design has the form of an online experiment, in which postgraduate students at the School of Social Sciences of the Hellenic Open University (H.O.U.) were invited to participate upon notification via email to their electronic academic account. Emails with the link of the questionnaire were sent en masse by the secretariat of the school, and the members of the research team did not have access to the students’ emails.

Students, constituting the population under study, generally represent a young population group, with a relatively lower risk of obtaining a serious COVID-19 infection compared to the elderly [13], a fact which could affect their degree of compliance with prevention measures [22]. As a matter of fact, an analysis of mobility trends has shown that individuals aged between 20 and 49 years old were responsible for sustaining the resurgence of the epidemic wave from the middle of 2020 in the USA, after the initial decrease in the number of cases [23]. In our case, postgraduate students in an open university were approached.

The sampling method is of the convenience type, and therefore, the present sample is not nationally representative. The free online questionnaire design platform “Google Forms” was used, and the sample was randomly divided into two groups depending on the month of birth (odd/even). Participants who stated an odd month made up the control group and were not exposed to any kind of message. On the contrary, those who stated an even month made up the treatment group and were exposed to an altruistically framed message, which consisted of a poster and an accompanying text. All of the other questions were common to both groups. It should also be noted that the ability of “Google Forms” to record participants’ emails was disabled, and participants had to give their consent in order to fill in the questionnaire anonymously, without receiving any kind of compensation.

As it is impossible to calculate the precise number of people invited, it is estimated that around 800 students received the email. The response collection period lasted 2 weeks (15–31 January 2022), during which a total of 425 responses were collected (N = 425). At that time, several personal protection measures were already in place, such as the mandatory use of masks both indoors and outdoors, maintaining distances, etc. [24]. Also, from mid-December 2021 to the end of the data collection period, there was a surge in COVID-19 cases, averaging 30,000 per day, with around 100 deaths per day [25].

The questionnaire was inspired by the research by Jordan et al. [5], Sasaki et al. [15], Dryhurst et al. [21], and Dohmen et al. [26]. Specifically, questions were selected and slightly modified, mainly by paraphrasing or using a different scale, in order to adapt them to the context of our study. Demographic and health information items were also included.

Based on Jordan et al. [5] and Sasaki et al. [15], compliance during the previous week with each of six prevention measures (“use of mask”, “frequent hand washing”, “avoidance of overcrowding”, “avoidance of touching face”, “adoption of contactless transactions”, and “avoidance of handshakes”) was measured through 11-point Likert scales from 0 to 10, where 0 indicates “strongly disagree”, 5 indicates “neither agree nor disagree”, and 10 indicates “strongly agree”. For statistical analysis purposes, a cumulative scale of the above six items from 0 to 60 was created, constituting the “Compliance” index. 

In the next section, according to Dohmen et al. [26], general and health risk attitudes were measured with two items with an 11-point Likert scale ranging from 0 to 10, where 0 indicates “completely unwilling to take risks in general” or “completely unwilling to take risks related to my health”, and 10 indicates “completely willing to take risks in general” or “completely willing to take risks related to my health”, respectively.

To measure COVID-19 risk perception, following the work by Dryhurst et al. [21] on the recent pandemic and that by van der Linden [27] on climate change, 5 items were used, based on 7-point Likert scales ranging from 1 to 7, with the labeling of reference points varying according to the question. The items assessed the extent to which subjects were concerned about the pandemic, their personal likelihood of infection, their family’s likelihood of infection, their estimate of the spread of the virus, and their understanding of the severity of the disease. The “Risk perception” index is a cumulative scale of the five items with a span from 5 to 35.

The intervention of the experiment involved a poster and an accompanying text about the coronavirus, with a question regarding the persuasiveness of the poster on a 7-point Likert scale ranging from 1 to 7 (1 = not persuasive at all, 7 = extremely persuasive), being asked immediately afterwards (see Figure 1).

Altruistic message text: “The COVID-19 virus continues to affect the global population. Recently, a new pandemic wave is taking place, making it clear that in the near future the health of your loved ones is under serious threat. It is recommended to follow directions regarding individual protection measures to prevent transmission and any serious illness or even death of your friends and relatives”.

Shortly afterwards, participants had to answer six questions about their intention to implement the same individual COVID-19 prevention measures in the near future. Following the example of Jordan et al. [5], the wording of the questions was slightly modified to emphasize that this time, it was a measure of the intention to implement the respective measures. A cumulative scale of six items ranging from 0 to 60 was created, constituting the “Intention” index. 

Finally, an attempt was made to fundamentally measure prosociality with a single question, through a 7-point Likert scale ranging from 1 to 7, and similarly to measure personal efficacy, by following the method used by Dryhurst et al. [21]. The questionnaire is presented in Appendix A. 

All factor scores were converted to a 0–100 scale using the following formula: 100 × (Xi − Min(X))/Range(X).

## 3. Results

In addition to the descriptive study, the statistical data analysis was focused mainly on statistical hypothesis testing and correlation analysis. After testing for normality with the Shapiro–Wilk test, it turned out that our data do not follow the normal distribution. Therefore, non-parametric tests (Mann–Whitney and Wilcoxon) were applied at a significance level of α = 0.05. Furthermore, in order to examine the relevance between qualitative variables, χ^2^ (chi-squared) independence tests were performed, while Cronbach’s alpha was calculated to assess the internal consistency reliability.

The sample size is N = 425 (N_control group_ = 208, N_treatment group_ = 217), with a median age of 40 years (24–59 years), and it consists of 102 male (24%) and 323 female (76%) students. The descriptive statistics of the qualitative medical profile variables are shown in Table 1, whereas the means and confidence intervals of the quantitative variables are shown in Table 2.

The Cronbach’s alpha is 0.703 for the six-item scale of “Compliance”, 0.802 for the corresponding scale of “Intention”, and 0.659 for the five-item scale of “Risk Perception”.

The change in compliance behavior of the participants in each group (intervention and control) was defined as the difference between the scales “intention to comply in the near future” and “compliance in the previous week”, which consist of the six precautionary measures (items). A Wilcoxon’s non-parametric test showed that there was a statistically significant difference between the intention to comply in the near future and compliance in the previous week, for both the treatment (*p* < 0.001) and the control (*p* < 0.001) groups. 

Next, the potential effect of the altruistic message (nudge) on the participants’ behaviors was assessed by the difference (if any) of the magnitudes of the changes in compliance behavior mentioned above in the intervention and control groups. In fact, as shown in Table 3, a Mann–Whitney non-parametric test showed that there was no statistically significant difference between the changes in compliance behavior in the two groups, i.e., the group exposed to nudging via the “altruistic message” and the control group. 

Therefore, Hypothesis 1, that exposure to an altruistic nudging message has a positive effect on the compliance behavior of individuals, is not supported by our analysis.

It should be noted that the Mann–Whitney tests showed that the two groups did not significantly differ in terms of age (*p* = 0.104), compliance (*p* = 0.728), risk perception (*p* = 0.186), general risk attitudes (*p* = 0.503), health risk attitudes (*p* = 0.896), prosociality (*p* = 0.130), and personal efficacy (*p* = 0.166).

In addition, the χ^2^ independence tests indicated that the partition method (month of birth: odd/even), was independent of gender (*p* = 0.248), personal history of COVID-19 (*p* = 0.702), family history of COVID-19 (*p* = 0.874), vulnerability to novel coronavirus (*p* = 0.346), and self-reported health status (*p* = 0.718).

Subsequently, the compliance, intention to comply, risk perception, general risk attitudes, health risk attitudes, prosociality, and personal efficacy variables were compared between subgroups based on sex, personal history of COVID-19, vulnerability to novel coronavirus, and corresponding vaccination status. 

The Mann–Whitney tests showed that the female students exhibited higher compliance (*p* = 0.001), intention to comply (*p* = 0.001), and risk perception (*p* = 0.001) than the male students, whereas the general risk attitudes (*p* = 0.678), health risk attitudes (*p* = 0.927), prosociality (*p* = 0.433), and personal efficacy (*p* = 0.101) did not differ between the males and females.

Between the students with and without a personal history of COVID-19, the Mann–Whitney tests showed that the former had lower intention (*p* = 0.026) and risk perception (*p* = 0.007) and higher health risk attitudes (*p* = 0.019), that is, higher risk-taking attitudes regarding health. Compliance (*p* = 0.153), general risk attitudes (*p* = 0.379), prosociality (*p* = 0.718), and personal efficacy (*p* = 0.132) did not differ significantly between these two subgroups.

Regarding vulnerability to the novel coronavirus, the Mann–Whitney tests showed that only risk perception (*p* = 0.003) varied significantly between the vulnerable and non-vulnerable students, with the former having a greater perception of risk from the disease. Compliance (*p* = 0.540), intention to comply (*p* = 0.248), general risk attitudes (*p* = 0.403), health risk attitudes (*p* = 0.470), prosociality (*p* = 0.308), and personal efficacy (*p* = 0.073) did not essentially differ.

As for the vaccination status against COVID-19, the Mann–Whitney tests suggested that the vaccinated students showed higher risk perception (*p* = 0.008) and personal efficacy (*p* = 0.040) than the non-vaccinated ones. Compliance (*p* = 0.292), intention to comply (*p* = 0.895), general risk attitudes (*p* = 0.174), health risk attitudes (*p* = 0.710), and prosociality (*p* = 0.158) did not vary substantially.

Furthermore, computed correlations showed that compliance was positively associated with risk perception (Spearman’s rho = 0.136, *p* = 0.005), intention to comply (Spearman’s rho = 0.788, *p* < 0.001), prosociality (Spearman’s rho = 0.200, *p* < 0.001), personal efficacy (Spearman’s rho = 0.334, *p* < 0.001), and age (Spearman’s rho = 0.382, *p* < 0.001), and negatively associated with general risk attitudes (Spearman’s rho = −0.135, *p* = 0.005) and health risk attitudes (Spearman’s rho = −0.263, *p* < 0.001). 

Therefore, Hypothesis 2, that compliance is associated with risk perception, is supported by our analysis.

## 4. Discussion

The COVID-19 pandemic is an important field of application of behavioral economics, as its tools can potentially motivate individuals to protect themselves from infection and therefore halt the spread of the virus [28]. The nudge studied in this research concerns the altruistic framing of an awareness message about the current pandemic and its effect on the intention to comply with preventive measures. Many interesting implications can be drawn.

A high rate of compliance with COVID-19 preventive measures was observed for the majority of the respondents. It seems that the mobilization of the media and the authorities led to a greater realization of the need to follow the instructions for self-protection, as was also seen in a survey conducted in Germany and Switzerland [29]. Furthermore, in accordance with the study by Dryhurst et al. [21], the measurement of COVID-19 risk perception and prosociality showed high values as well, and the two variables were positively—yet weakly in our case—correlated with compliance. In fact, Kanellopoulou et al. [30] attributed the great willingness to implement preventive measures against COVID-19 by Greek citizens to the high degree of risk perception. This is expected, since people are more likely to abide by preventive measures if they consider the virus a serious threat to themselves and their beloved ones. As for the factor of prosociality, several researchers agree that the degree of pro-social behavior displayed by individuals acts beneficially on compliance and the intention to comply with prevention measures, helping, in this way, to decelerate the spread of the virus [17,31].

Regarding the effect of the altruistic message, the documented difference between the intention to comply with preventive measures against COVID-19 and previous compliance did not vary significantly between the treatment and control groups. It follows that nudging via an altruistic message had no effect on behavior. These results contradict the finding of increased intentions through prosocial incentives by Sasaki et al. [15], and are in line with those of Jordan et al. [5], who found no reinforcing action of altruistic framing at the follow-up, despite the initial encouraging results. Presumably, motivational messages cannot bring any additional benefit due to the fact that the majority of participants already appear to be quite compliant with the measures [32]. Favero and Pedersen [18] reported that messages highlighting the value of social distancing in reducing the spread of the novel coronavirus are expected to have better results in the early stages of the pandemic. In addition, Nese et al. [33] showed that a longer duration of imposed restrictive measures is associated with greater relaxation in observed preventive behaviors. As we go through the second year of this complex crisis, it is perhaps natural to observe the phenomenon of fatigue in a significant part of the Greek society due to the continuous communication of messages by the government prompting preventive behavior, and thus making their further promotion ineffective. Finally, we cannot preclude the possibility that the nudge itself was in fact rather small, and the specific message was not stimulating enough, in a period when individuals were being bombarded with messages about COVID-19 measures. In this respect, a more powerful nudge than the brief awareness message employed in the present study might be required in order to enhance compliance, especially in later phases of a pandemic.

The effects of demographic characteristics and medical profiles of the respondents on particular variables of interest were also examined. To begin with, the female students showed higher compliance, intention to comply, and risk perception, while they appeared to share the same general and health risk attitudes, prosociality, and personal efficacy with the male students. Students with a personal history of COVID-19 are willing to take more health risks and have a lower risk perception and intention to comply with preventive measures. A UK health system survey [34] found that, due to the high prevalence of asymptomatic disease among citizens, those who were ill but not hospitalized showed a lower risk perception of the disease. These findings, along with ours, are not in line with the research findings of Dryhurst et al. [21], which probably referred to a more lethal variant of the virus and a period when vaccines were not readily available to the public.

Furthermore, the participants who are vulnerable to the novel coronavirus showed greater risk perception, a fact that is not accompanied by a higher intention to abide by individual prevention measures. According to Ramkissoon [35], people who belong to vulnerable groups are already compliant with preventive measures against infections and thus can more easily follow the suggested behaviors, as instructed by specialists.

Finally, it is worth mentioning that the vaccinated students showed greater risk perception and personal efficacy. According to Wright et al. [36], individuals who get vaccinated against COVID-19 do not show less compliance with protective measures, implying a high risk perception.

## 5. Conclusions

At the specific stage of the pandemic in which the study was conducted, the altruistic framing used was not found capable of further strengthening the intention to adopt preventive behaviors against COVID-19. Moreover, as expected, we found that a higher risk perception is associated with higher compliance. To our knowledge, this was the first study conducted in the middle of the pandemic and, along with prior research, allows for a more comprehensive assessment of nudging effectiveness.

Regarding the theoretical and policy implications of the study, it is apparent that the effectiveness of nudges should not be taken for granted. The specific circumstances of a pandemic should be assessed prior to any practical implementation. If a pandemic is not in its initial phase, then the use of powerful nudges might be necessary, especially if a high level of compliance exists. This, for instance, might be the case where younger individuals have a lower risk of severe infection from a virus compared to older people, which, in turn, might affect their compliance with the recommended preventive measures [13,22]. It is recommended that, prior to any implementation of nudging, an assessment be carried out of the compliance level of the population of interest. If available evidence exists that documents the effectiveness of specific nudges that are strong enough to affect certain groups of the population, then they should be applied, ideally those that are the most cost-effective. If there is a lack of relevant information, perhaps some form of pilot testing would be preferable before actual implementation on a population basis.

This study has some limitations. The sample consists of postgraduate students in a single open university. Although their median age is in fact only 5 years lower than the Greek population’s respective value, it consists of more educated persons than average. It is also a student population in Greece, which might differ in many respects from students in other countries (e.g., due to different levels of prosociality, risk perception, risk attitudes, etc.). Moreover, the present research measured the self-reported intention to comply with preventive measures rather than actual behavior, which might not always coincide [37]. And the specific nudge applied via a brief message might not have been powerful enough, given the phase of the pandemic and the bombardment of messages already in effect through the mass and social media. 

Future research should employ larger targeted or representative samples of the general population in order to assess the comparative effectiveness of nudges of different magnitudes (e.g., with more or less powerful wording or messages), at different pandemic periods (e.g., during different pandemic waves), in different populations (e.g., students in different countries, or the elderly and minority groups that are more vulnerable [38]), and implemented via different communication channels (e.g., e-mails, smartphones, TV, radio, and social media) in order to enrich our knowledge and guide policy making. Towards this end, simulation models that explore the cost-effectiveness of different nudging scenarios under different circumstances will also prove useful. Such research will allow public health officials to make evidence-informed decision making to increase social welfare. 

## Figures and Tables

**Figure 1 ijerph-21-00031-f001:**
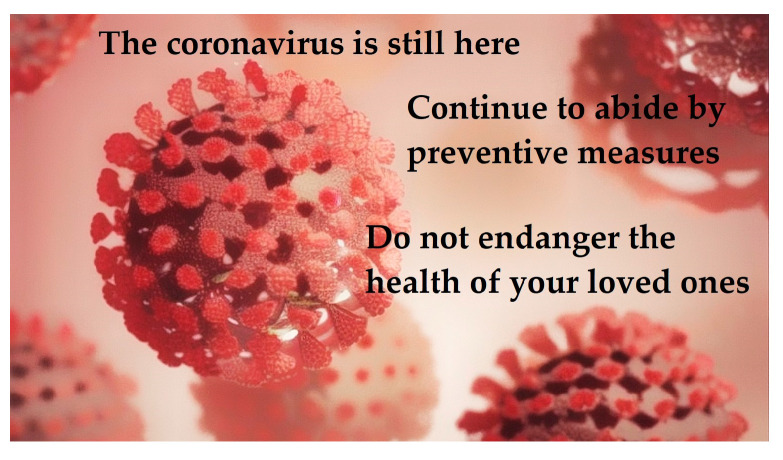
Altruistic message poster.

**Table 1 ijerph-21-00031-t001:** Participant’s medical profiles (N = 425).

Nominal Variables	Category	f	%
Personal history of COVID-19	Positive	121	28.5
	Negative	304	71.5
Family history of COVID-19	Positive	219	51.5
	Negative	206	48.5
Vulnerability to novel coronavirus	Yes	45	10.6
	No	372	87.5
	I don’t know	8	1.9
Self-assessment of health status	Very bad	1	0.2
	Bad	2	0.5
	Moderate	18	4.2
	Good	96	22.6
	Very good	193	45.4
	Excellent	115	27.1
COVID-19 vaccination status	Vaccinated	408	96.0
	Non-vaccinated	17	4.0

**Table 2 ijerph-21-00031-t002:** Means, standard deviations, and confidence intervals for items and scales (N = 425).

Item/Scale	Unadjusted Mean *	Mean	S.D.	95% CI Lower	95% CI Upper
Use of mask **	9.42	94.24	11.85	93.11	95.37
Frequent handwashing **	9.33	93.32	11.60	92.21	94.42
Avoidance of overcrowding **	8.74	87.41	18.89	85.61	89.21
Avoidance of touching face mucosa **	8.28	82.82	20.83	80.84	84.81
Adoption of contactless transactions **	8.45	84.52	19.81	82.63	86.41
Avoidance of handshakes **	8.84	88.38	17.39	86.72	90.03
Worry about the pandemic	4.57	59.57	25.72	57.12	62.02
Personal chance of infection	5.32	72.08	24.14	69.78	74.38
Familial chance of infection	5.66	77.61	20.31	75.67	79.54
Assessment of virus spread	6.39	89.76	14.22	88.41	91.12
Understanding disease severity	6.41	90.24	15.42	88.76	91.71
Compliance ***	53.07	88.45	10.87	87.41	89.48
Risk perception ***	28.36	77.85	13.32	76.58	79.12
Intention ***	54.27	90.44	11.08	89.39	91.45
General risk attitudes	5.44	54.35	26.70	51.80	56.90
Health risk attitudes	3.10	30.99	28.43	28.28	33.70
Prosociality	5.93	82.16	17.84	80.46	83.86
Personal efficacy	6.26	87.73	16.08	86.19	89.26

Note: * denotes the mean scores of the items or scales prior to their conversion on 0–100 scores. ** denotes recent compliance with each preventive measure (the items regarding intentions to comply with each measure separately are not shown). *** denotes scales rather than individual questionnaire items.

**Table 3 ijerph-21-00031-t003:** Recent compliance and intention to comply with prevention measures in the near future in the treatment and control groups.

Variable	Control Group	Treatment Group	Z	*p*-Value
Compliance (A)	88.74	88.16	−0.348	0.728
Intention (Β) Difference (Β)—(A)	90.52 1.78	90.37 2.20	−0.284 −0.022	0.776 0.982

## Data Availability

The data that support the findings of this study are available from the corresponding author, V.A., upon reasonable request.

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
