# Peer review of "The Effect of Nudging on Compliance with Individual Prevention Measures against COVID-19: An Online Experiment on Greek University Students"

_ijerph, 2023, doi:10.3390/ijerph21010031_

Round 1

Reviewer 1 Report

Comments and Suggestions for Authors

The subject under study is certainly important, especially in the historical period we are experiencing. The article presents interesting results, but it is nevertheless believed that, given the organization of the contents and the description of the same, the manuscript cannot be published in its current form, especially for its local impact. I would like to encourage authors to consider several issues to be improved.

 Introduction: The introduction (line90-112) is too long and confusing. It should be shortened highlighting the objectives of this section. A good introduction clearly sets out what is missing/wrong with the existing theories and why a modification is needed. Discussion of research hypotheses should be made point by point. What is the possible international contribution of the study to the literature? What are the implications of the study? The objectives should be better explained at the end of the section. It should state what the study contributions are and how this ads to the literature. Moreover, it should clearly state the research question. Behavioral economics seems to be a key concept in the article. However, this concept was only briefly mentioned in the study and was not deeply integrated with the study.

 Materials and Methods: The presentation of measurement index is not clear. For example, measurement categories and corresponding literature should be presented more clearly (line 146-147). For another example, the Likert scale in this study uses different evaluation criteria, and what are the reference criteria? For example, what is the basis for the value of 0? The Likert scale conversion formula lacks reference labeling.

 Discussion:It is recommended to integrate the discussion of the results in terms of the differences exhibited by the demographic characteristics of the conclusion section. It is suggested that the language be refined to highlight key points.

 Conclusions:The conclusion is not concrete. This section is lengthy and verbose, and it is recommended that the main findings be summarized in concise language. The discussion of the results in terms of differences in demographic characteristics is proposed to be moved to section 4. Discussion.

 Authors should clearly indicate the research contribution (theoretical, practical), research limitations, and methods or strategies for improvement at the end of the article.

Author Response

We greatly appreciate and thank the reviewer for the thorough review of our manuscript and the insightful comments.

REVIEWER #1

Comment #1 (Introduction): The introduction (line 90-112) is too long and confusing. It should be shortened highlighting the objectives of this section. A good introduction clearly sets out what is missing/wrong with the existing theories and why a modification is needed. Discussion of research hypotheses should be made point by point. What is the possible international contribution of the study to the literature? What are the implications of the study? The objectives should be better explained at the end of the section. It should state what the study contributions are and how this adds to the literature. Moreover, it should clearly state the research question. Behavioral economics seems to be a key concept in the article. However, this concept was only briefly mentioned in the study and was not deeply integrated with the study.

Authors’ reply: We have shortened the introduction by about 250 words and made it clearer. We now highlight the gap in the literature that the study addresses, discuss research hypotheses point by point and state the study’s contribution to the literature. We follow the relevant empirical literature regarding the discussion of nudging within our study. We added a point on how our findings can be exploited from an economic perspective (lines 95-96 and 359-362 in the revised manuscript).

Comment #2 (Materials and Methods):  The presentation of measurement index is not clear. For example, measurement categories and corresponding literature should be presented more clearly (line 146-147). For another example, the Likert scale in this study uses different evaluation criteria, and what are the reference criteria? For example, what is the basis for the value of 0? The Likert scale conversion formula lacks reference labeling.

Authors’ reply: We have presented the measurement index and categories more clearly (lines 135-175). Given the on-going research on reference labelling we have not used labels for all intermediate points in the scales. The conversion was made for a more homogeneous presentation of scale scores. However, for the shake of completeness we have now also included the unconverted scores in Table 2, with values as measured in terms of the original scale categories of the study. The full questionnaire is presented in the Appendix.

Comment #3 (Discussion): It is recommended to integrate the discussion of the results in terms of the differences exhibited by the demographic characteristics of the conclusion section. It is suggested that the language be refined to highlight key points.

Authors’ reply: Discussion of results in terms of the differences exhibited by the demographic characteristics was removed from the Conclusion Section and moved into the Discussion Section. The language has been refined.

Comment #4 (Conclusions): The conclusion is not concrete. This section is lengthy and verbose, and it is recommended that the main findings be summarized in concise language. The discussion of the results in terms of differences in demographic characteristics is proposed to be moved to section 4. Discussion Authors should clearly indicate the research contribution (theoretical, practical), research limitations, and methods or strategies for improvement at the end of the article.

Authors’ reply: As noted, discussion of results in relation to the sample’s demographics has been removed from the Conclusions Section. In addition, we have made the section shorter and more concrete by following the reviewer’s advice. It now contains clear presentation of findings, implications, limitations and suggestions for future research.

Reviewer 2 Report

Comments and Suggestions for Authors

This is a worthwhile paper and I hope it gets published. The design of the study is correct, with appropriate statistics and control group. My only reservation is that the "nudge" was so small. It seems a minor message, sent out during a time when people everywhere were being bombarded with messages about Covid measures, so it is rather hard to attribute effects to this message alone. Somewhere the authors need to mention this point in their discussion.

There are two parts that I do not fully understand, so please supply further explanations:

Line 319: Apparently, the altruistic message was in a sense subjected to an endurance test, as it was implemented during the second year of the pandemic in Greece.

(what do you mean?)

Line 328: the observed difference between intention and compliance in both groups does not seem to be a significant improvement-

(please check meaning here - this is not correctly phrased if you mean that the difference does not indicate a significant improvement)

Comments on the Quality of English Language

A few minor points. The English is mainly correct (but please use "reported" rather than "referred" when you mean "said"). The discursive parts (intro and discussion) might benefit from a quick revision to strengthen the arguments and boost fluency.

Author Response

We greatly appreciate and thank the reviewer for the thorough review of our manuscript and the insightful comments.

REVIEWER #2

Comment #1: This is a worthwhile paper and I hope it gets published. The design of the study is correct, with appropriate statistics and control group. My only reservation is that the "nudge" was so small. It seems a minor message, sent out during a time when people everywhere were being bombarded with messages about Covid measures, so it is rather hard to attribute effects to this message alone. Somewhere the authors need to mention this point in their discussion.

Authors’ reply: We integrated this remark both in the Discussion (lines 295-300) and the Conclusion Section (as a limitation, see lines 350-352).

Comment #2: Line 319: Apparently, the altruistic message was in a sense subjected to an endurance test, as it was implemented during the second year of the pandemic in Greece (what do you mean?)

Authors’ reply: We have deleted the sentence as it was indeed confusing.

Comment #3: Line 328: the observed difference between intention and compliance in both groups does not seem to be a significant improvement - (please check meaning here - this is not correctly phrased if you mean that the difference does not indicate a significant improvement).

Authors’ reply: We have made language corrections not only in the suggested sentence but in the entire Discussion Section in the revised manuscript).

Comments #4: A few minor points. The English is mainly correct (but please use "reported" rather than "referred" when you mean "said"). The discursive parts (intro and discussion) might benefit from a quick revision to strengthen the arguments and boost fluency.

Authors’ reply: We have corrected the word “referred” and made a revision of the Introduction, Discussion and Conclusion sections to make them more concrete and boost fluency.

Reviewer 3 Report

Comments and Suggestions for Authors

The paper "The effect of nudging on compliance with individual prevention measures against COVID-19: An online experiment on Greek university students" discusses the effect of nudging on compliance with individual prevention measures against COVID-19 among Greek university students. 

The study evaluates the impact of an altruistically framed awareness message about the novel coronavirus on the intention to comply with infection prevention measures. 

The paper needs a careful revision before it can be published. 

Below, I present some points for improvement:

  1. The paper states that "The questionnaire was based on the research of Jordan et al. [5], Sasaki et al. [18], Dry- 146 Hurst et al. [17] and Dohmen et al. [26] ."  Explain in more detail how the questionnaire is constructed. Which questions are from each questionnaire? Are the questions original and inspired by these papers, or are the questions the same? Clarify these issues to help readers. 

2. The paper finds that nudging interventions should be evaluated before implementation, considering factors like timing, target groups, and means of communication. This message is quite important and could be explored further. 

3. Intention is different from action. The framing induces a different response but on intentions. What are the limitations of the study?

4. How do other researchers can explore these results and add to this research? Are the results specific to Greek students? Or should we expect students from other parts of the world to present similar response patterns? Add some suggestions for further research.

Overall, I enjoyed reading the paper, and it is an excellent contribution to a very relevant topic: using nudges to improve public health policies.

Author Response

We greatly appreciate and thank the reviewer for the thorough review of our manuscript and the insightful comments.

REVIEWER #3

Comment #1: The paper states that "The questionnaire was based on the research of Jordan et al. [5], Sasaki et al. [18], Dry-Hurst et al. [17] and Dohmen et al. [26] . "  Explain in more detail how the questionnaire is constructed. Which questions are from each questionnaire? Are the questions original and inspired by these papers, or are the questions the same? Clarify these issues to help readers.

Authors’ reply: In the revised manuscript we explain in more detail how the questionnaire was constructed (see lines 132-155, 167-169).

Comment #2: The paper finds that nudging interventions should be evaluated before implementation, considering factors like timing, target groups, and means of communication. This message is quite important and could be explored further.

Authors’ reply: We revised the Conclusion Section completely as requested by another reviewer and made it shorter. We now explain how we view the above factors in terms of comparative analyses as well as in terms of inputs into a simulation model that would assess the cost-effectiveness of different scenarios based on timing, means of communication etc. (see lines 353-362).

Comment #3: Intention is different from action. The framing induces a different response but on intentions. What are the limitations of the study?

Authors’ reply: We have added this into the Conclusion Section (see, limitations, lines 348-349).

Comment #4: How do other researchers can explore these results and add to this research? Are the results specific to Greek students? Or should we expect students from other parts of the world to present similar response patterns? Add some suggestions for further research.

Authors’ reply: We have made the Conclusions Section more concrete. We have added suggestions for future research, including comparisons of nudges between students in different countries (lines 343-362).

Round 2

Reviewer 1 Report

Comments and Suggestions for Authors

The article is acceptable this time.

Reviewer 3 Report

Comments and Suggestions for Authors

The paper seems to have improved.